

# Structure and characteristics of the plant-frugivore bird network from the Guilin Botanical Garden

Guohai Wang[1,2], Yang Huang[2], Wei Yao[2], Qiuchan Huang[1], Yongping Huang[3], Lijuan Wei[3] and Qihai Zhou[2]

[1] College of Chemistry and Bioengineering, Guangxi Normal University for Nationalities, Chongzuo, Guangxi, China
[2] Key Laboratory of Ecology of Rare and Endangered Species and Environmental Protection, Ministry of Education; Guangxi Key Laboratory of Rare and Endangered Animal Ecology, Guangxi Normal University, Guilin, Guangxi, China
[3] College of Mathematics, Physics and Electronic Information Engineering, Guangxi Normal University for Nationalities, Chongzuo, Guangxi, China

Corresponding authors
Lijuan Wei, 156186148@qq.com
Qihai Zhou, zhouqh@ioz.ac.cn

## ABSTRACT

The interaction between plants and frugivores is crucial to ecosystem function and community diversity. However, little is known about the interaction between plants and frugivorous bird species in urban green spaces. We observed interactions between plants and frugivorous birds in the Guilin Botanical Garden for one year and determined the structure and characteristics of the interaction network. We also analyzed the impact of species traits on their network roles. Interactions between 14 frugivorous birds and 13 fruit plant species were recorded in the study area. Autumn interactions comprised 38.79% of the overall network, and winter interactions comprised 33.15%. The modularity ($Q$, $z$-score) of the network was higher in autumn; the weighted nestedness ($wNODF$, $z$-score) and interaction evenness ($E_2$, $z$-score) of the network were higher in winter; the connectance ($C$, $z$-score) and interaction diversity ($z$-score) of the network were higher in spring; and the specialization ($H_2'$, $z$-score) of the network was higher in summer. The observed network showed lower $C$, lower interaction $H_2$, lower $E_2$, lower $wNODF$, higher $H_2'$ and higher $Q$ when compared to the random networks. The bird species most important to network stability were *Hemixos castanonotus*, *Parus venustulus*, and *Pycnonotus sinensis*. The most important plant species were *Alocasia macrorrhiza*, *Cinnamomum camphora*, and *Machilus nanmu*. Of all the bird and plant traits included in this study, only plant color had a significant impact on species strength, with black fruit having a higher species strength. Our results suggest that interaction networks in urban green spaces can be temporally complex and variable and that a network approach can be an important monitoring tool for detecting the status of crucial ecosystem functions.

## INTRODUCTION

The relationship between frugivores and plants directly impacts the ecosystem, and the stability of this relationship affects plant population dynamics, community structure, biodiversity maintenance, and degraded ecosystems regeneration (*García, Donoso & Rodríguez-Pérez, 2018*; *Rumeu et al., 2020*). Birds are important dispersal vectors for fleshy-fruited plants because of their high diversity, mobility, diverse habitat selection (*Carlo & Morales, 2016*; *Camargo et al., 2020*), and specific body traits that allow them to concurrently consume a variety of fruit species (*Muñoz et al., 2017*). The foraging behavior of frugivorous birds on fleshy-fruited plants can form a complex interaction network (*Saavedra et al., 2014*). Unraveling the structure and dynamics of this network can provide novel insights into co-evolution theory (*Schleuning, Fründ & García, 2015*), and can be used to reveal ecological patterns and to plan conservation efforts (*Ramos-Robles, Andresen & Díaz-Castelazo, 2016*; *Rocha-Filho et al., 2022*).

Rapid urbanization has significantly altered many ecosystems, resulting in the loss or fragmentation of natural habitats, and the creation of insular habitats (*Kiers et al., 2010*). Changes in landscape configuration and composition affect the movement and community diversity of bird populations (*Pena et al., 2017*), and ultimately reduce the stability of the interaction network structure and ecosystem service function which is detrimental to ecological and evolutionary systems (*Harrison & Winfree, 2015*; *Guenat et al., 2019*). Urbanization reduces bird species richness and increases interaction evenness, negatively affecting the stability of the plant-bird interaction network (*Schneiberg et al., 2020*). Urban green spaces, such as patched native vegetation and artificially managed parks, are important habitats for urban birds and play a crucial role in maintaining species diversity in urban ecosystems (*Daniels et al., 2020*; *Zhang et al., 2022*). Urban green spaces comprise a range of garden plants, which not only provide sufficient food resources and suitable alternative habitats for birds, but are also connected through the dispersal behavior of frugivorous birds (*Silva et al., 2015*; *He et al., 2022*). For instance, 11 bird species foraged 15 fleshy-fruited plants and formed 33 network links in an urban park in Portugal (*Cruz et al., 2013*).

The interaction network between birds and plants is asymmetric (*Sebastián-González, 2017*), indicating that different species have different functional roles in the interaction network (*Montoya-Arango, Acevedo-Quintero & Parra, 2019*). The functional role of a species is most accurately described by the functional traits of its interaction partners; some birds can form an interaction relationship with a variety of plants, while others have a reduced number of interacting partners (*Dehling et al., 2016*; *Bender et al., 2017*). Previous studies have shown that the role of a species in an interaction network is associated with specific species traits, such as foraging behavior, bird body size, fruit size, and fruit resource availability (*Vizentin-Bugoni et al., 2021*). For example, large-bodied birds tend to have broader diets and larger home ranges (*Li et al., 2018*). Bird species richness also strongly influences both interaction frequency and habitat; thus, bird species with more distinct traits perform more unique functional roles in the interaction network (*Poisot, Stouffer & Gravel, 2015*; *Rumeu et al., 2020*). Therefore, identifying the species traits that

are related to the different roles played by different species in the network may be important for understanding their impact on the overall interaction network (*Dehling et al., 2016*; *Sebastián-González, 2017*).

Many botanical gardens around the world, due to their higher plant diversity have fleshy fruits throughout or at least during a larger proportion of the year in comparison with the native flora, especially outside tropical and sub-tropical climates. However, little information is available regarding the structure and characteristics of the interaction network between frugivorous birds and fruit plants in botanical gardens. In this study, we used the Guilin Botanical Garden as an example to study the structure and characteristics of the interaction network between frugivorous birds and fruit plants. We aimed to explore the role of urban green space in maintaining species interaction networks and ecosystem functions, and focused on the following research questions: (1) whether seasonal variation affects the interaction network between frugivorous birds and plant species and (2) whether specific species traits affect the functional roles of those species in the interaction networks. It was predicted that (i) the network structure has seasonal differences, and (ii) the role of different species in the interaction network is affected by species traits.

## METHODS

### Study site

We conducted this study in the Guilin Botanical Garden (25°04′N, 110°17′E), Guangxi Zhuang Autonomous Region, Southwest China (Fig. 1). It was established in 1958 and covers an area of approximately 73 ha at an altitude of 180–300 m. The mid-subtropical monsoon dominates this region's climate, with average annual temperatures of 19.2 °C, and minimum and maximum temperatures of −4.2 °C in January and 36 °C in July, respectively. The annual mean relative humidity of the area is greater than 78%, with an average of 1,800 mm of precipitation per year according to *Lu et al. (2020)*. The local vegetation consists of middle subtropical evergreen and deciduous broad-leaved mixed forests and the fleshy fruit plants are composed mostly of native trees, such as *Cinnamomum camphora*, *Machilus thunbergii*, and *Ficus concinna*. Their main frugivores are Chinese Bulbul (*Pycnonotus sinensis*), Red-whiskered Bulbul (*Pycnonotus jocosus*), and Japanese White-eye (*Zosterops japonicus*), which are the resident species.

### Experimental design
#### Plant-frugivore bird network
Based on the distribution characteristics of fruit plants, we established four transects (2–3 km for each transect) in the survey area to observe the foraging behavior of frugivorous birds from September 2020 to August 2021. Bird foraging sampling was performed using 8–10 × 42 zoom binoculars (Bosma, Boguan Photoelectric Technology Co., Ltd Guangzhou, China) during two foraging periods: 7:00–10:00 and 14:00–17:00. Once birds were found foraging on plant fruits, the observation was conducted for 30 min to obtain sufficient foraging records, and the species of both the birds and plants foraged, as well as the number of fruits, number of birds per visit, and foraging time were recorded. If frugivorous birds visited the trees in conspecific flocks, the habits of one randomly selected

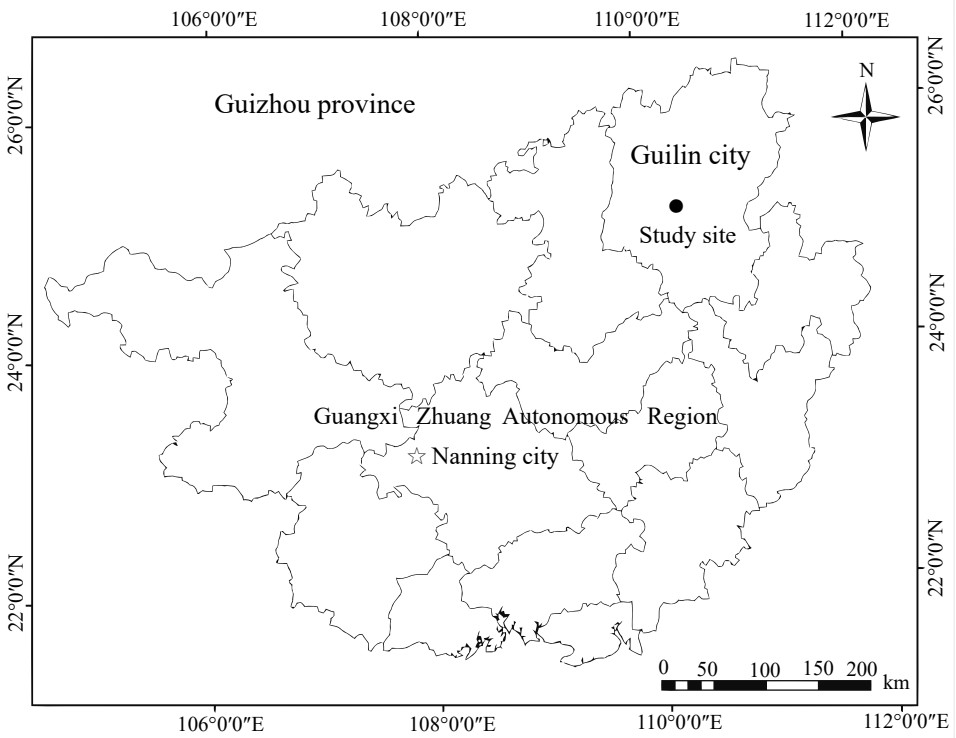

**Figure 1** Location map of the Guilin Botanical Garden, China.

bird were recorded and assumed to be characteristic of the entire feeding flock (*Breitbach et al., 2010*). The observation frequency was at least 8 days for each transect every month. Field observations revealed, that some birds foraged the fruits by pecking, which made it difficult to record a realistic foraging quantity. In order to prevent foraging style from impacting the interaction network results (*Jordano, 2016*; *Zhang et al., 2022*), this study used frequency of bird visits to plant species to build the interaction networks between frugivorous birds and fruits plants.

## Fruit plant and frugivorous bird traits

The traits of all fruit plants and frugivorous birds relevant to their roles in the interaction network were recorded. Early observations found that the fruit plants that birds forage on native plants, and the plant traits include fruit diameter (mm), fruit length (mm), fruit mass (g), fruit volume (mm$^3$), and fruit color. Vernier calipers and an electronic balance were used to measure the length, diameter, and mass of 20 fruits from one to three trees per species. The fruit volume was calculated using the ellipsoid volume calculation: v = 4/3($\pi \times$l/ 2$\times$d/ 2$\times$d/2) according to (*Zhang et al., 2022*). The bird traits recorded included bill width (mm), body length (mm), body mass (g), wing length (mm), and wing loading. Wing loading =body mass/ 2$\times$wing length was used to reflect the movement ability of the birds, movement capacity lessens as wing loading value increases (*Ruggera et al., 2016*; *Camargo et al., 2020*). The bird traits used in this study were obtained from a field guide to

the birds of China (*Mackinnon & Philipp's, 2000*) and A Handbook of the Birds of China (*Zhao, 2001*).

## Data analysis
### Network structure description
Foraging frequency was used to analyze seasonal differences in the plant-frugivore network, and the entire year was split into four seasons: spring (March, April, and May), summer (June, July, and August), autumn (September, October, and November), and winter (December, January, and February). The weighted interaction network's structure was analyzed at the function "networklevel" using the "bipartite" package in R (*Zhang et al., 2022*) based on the following parameters: (1) connectance (C), which varies from 0 (no interactions) to 1 (all species connected to each other), is the proportion of realized interactions in relation to the total interactions possible in the network, based on formula $C = n/(b \times p)$, where $n$ is the number of observed interactions, $b$ is the number of bird species, and $p$ is the number of plant species detected in the study (*Blüthgen et al., 2008*); (2) weighted nestedness (*wNODF*), which measures the extent to which species with few interactions are connected to species that are highly connected and is also related to network stability (*Bascompte et al., 2003*); (3) specialization ($H_2\prime$), which measures the overall level of specialization in a network, or whether a species in a network has shared or unique interaction partners (*Blüthgen, Menzel & Blüthgen, 2006*); (4) interaction diversity ($H_2$) is calculated using the Shannon-Wiener index based on interaction frequency, which reflects if the links are strong (high interaction frequencies) or weak (low interaction frequencies) (*Blüthgen, 2010*); (5) interaction evenness ($E_2$), which measures heterogeneity in the distribution of interactions across species in the network, with high values indicating more even distribution (*Dormann et al., 2009*); and (6) modularity (Q), which varies from 0 (no modularity) to 1 (network is organized into modules), and indicates whether or not the network is organized into distinguishable modules or compartments, or whether there is a group or subset of species that interact with each other more than would be expected by chance (*Barber, 2007*).

The null model function ("null model 1") was used to randomize the plant-frugivore interactions and compare the structural differences between the observed network and the null model (1,000 iterations). Under a simplistic null hypothesis, randomizations can determine which nodes (species) interact with one another and the strength of these interactions. Randomization can also determine whether the frequency of interactions between consumers and resources is a result of the relative abundance of potential resources (*Vaughan et al., 2018*). "Null model 1" can randomize the interactions among species while maintaining network size and the proportion of realized ecological interactions among all potential interactions in a network (*Dormann et al., 2009*).

Due to differences in species richness and heterogeneity of interactions, the values of the connectance, weighted nestedness, specialization, interaction diversity, interaction evenness, and modularity metrics were subjected to standardization of effect size when comparing the changes of network parameters in different seasons through $Z$-score: $Z = (\text{Obs} - \text{Exp}_{null(1...n)})/\text{Sd}_{null}$, where Obs is the observed value, $\text{Exp}_{null(1...n)}$ and $\text{Sd}_{null}$ are the

mean value and standard deviation of 1,000 randomizations derived from the null model (*Ulrich & Gotelli, 2007*).

## Network roles

The network roles of plants and frugivorous birds were characterized at the function "specieslevel" with the "bipartite" package in R (*Vollstädt et al., 2018*) based on the following parameters: (1) species degree, which is the percentage of potential partners a species interacts with and relates to how significant the species is to the stability and cohesiveness of the overall network of connected species (*Bascompte, Jordano & Olesen, 2006*); (2) species strength, which depicts the relative importance of a plant species for the assemblage of bird species (*Bascompte & Jordano, 2007*); (3) partner diversity, which represents the diversity of interaction partners for each species and is a quantitative analog to the qualitative species measurement or the richness of interaction partners (*Blüthgen, Menzel & Blüthgen, 2006*); (4) effective partners, which explains the range of various partners a species interacts with in a given network (*Bersier, Banašek-Richter & Cattin, 2002*); and (5) specialization, which assesses the degree to which a species deviates from a random sample of interaction partners, presuming that all species interact in accordance to their total frequencies (*Blüthgen, Menzel & Blüthgen, 2006*). The importance of each species to the stability of the network (that is, their ability to recover from small perturbations) was determined by calculating the contribution of each species to nestedness (CN), with species with positive values having more important roles than those with negative values (*Saavedra et al., 2011*).

Pearson's correlations were used to analyze the relationships between the interaction connections proportion to the total interaction connections of the network in different seasons and the number of frugivorous birds and plants. Generalized linear models (GLMs; "lme4" package, version 4.2.0, *R Core Team, 2019*) were used to estimate the effect of species traits on their network role, with network parameters (species degree, species strength, partner diversity, effective partners, and specialization) as the dependent variable and bird traits (body mass, body length, bill length, wing length, and wing-loading) and plant traits (fruit mass, fruit length, fruit diameter, fruit volume, and fruit color) as the independent variables. A statistically significant difference was defined as $P < 0.05$.

## RESULTS

### Yearly and seasonal changes of the interaction network

The interaction network included 14 bird species (two order, seven families) and 13 plant species (nine order, 10 families), with a total of 2,235 interactions observed throughout the year (Fig. 2). Each plant species interacted with $7.08 \pm 1.01$ (Mean ±SD) birds, and each bird species interacted with $6.13 \pm 1.15$ plant species (Fig. 2). All these plants are native to China, and 12 of the bird species are classified as resident bird species. *Turdus cardis* and *T. hortulorum* are classified as migrant bird species, and 13 of the 14 species of fruit-foraging birds are passeriformes, which accounted for 98.26% of the total observed interactions (Fig. 2). *Cayratia japonica*, with 724 interactions (32.39%), was the plant most commonly
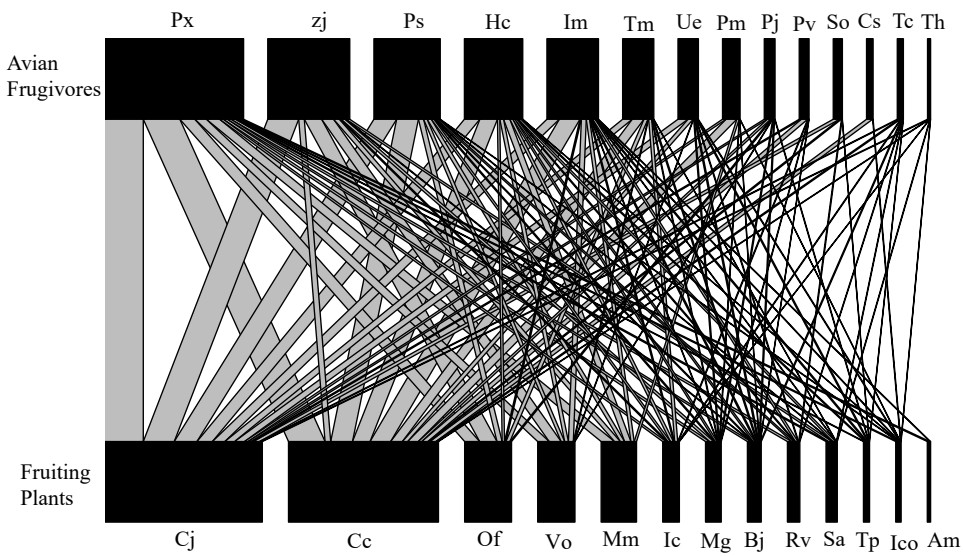

**Figure 2** Bipartite network diagram between plants (13 spp.) and birds (14 spp.) in the Guilin Botanical Garden, China.

consumed by birds, and *Pycnonotus xanthorrhous* was the most frequently recorded bird species, with 644 total interactions (28.81%, Fig. 2).

The observed plant-frugivore network showed lower connectance ($C = 0.604$), lower interaction diversity ($H_2 = 3.909$), lower interaction evenness ($E_2 = 0.751$), lower weighted nestedness ($wNODF = 56.405$), higher specialization ($H' = 0.091$), and higher modularity ($Q = 0.124$) when compared to the random networks produced by the null model ($N = 1,000$; Fig. 3). These findings showed that the observed network had fewer realized connections, fewer species with higher foraging dependence, and lower interaction frequencies than the random network.

The proportion of the interaction connections to the total connections of the network was different according to season (Fig. 4). Interaction connections in spring comprised 13.38% of the annual network connections, 14.68% in summer, 38.79% in autumn, and 33.15% in winter. The modularity ($z$-score) of the network was higher in autumn; the weighted nestedness ($z$-score) and interaction evenness ($z$-score) of the network were higher in winter; the connectance ($z$-score) and interaction diversity ($z$-score) of the network were higher in spring; and the specialization ($z$-score) of the network was higher in summer (Table 1). The proportion of interaction connections to the total interaction connections of the network in different seasons has a significant positive correlation with the number of frugivorous birds ($R^2 = 0.954$, $P = 0.023$). There was no significant correlation between the proportion of interaction connections to the total interaction connections of the network in different seasons and the number of plants ($R^2 = 0.297$, $P = 0.455$).

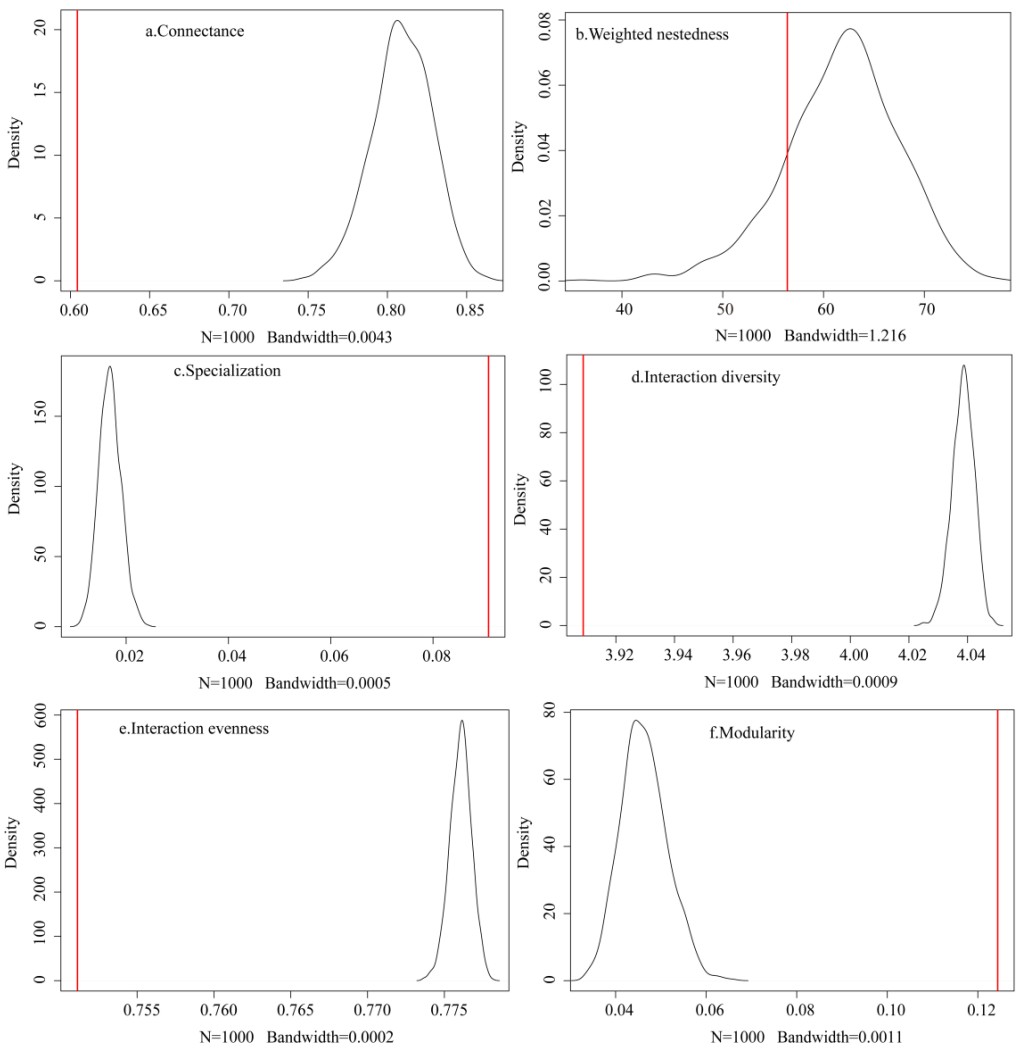

**Figure 3** **(A–F) Comparison between the observed network and the networks generated by the null model.**

### Effect of species traits on the functional roles of each species

Different species have different functional roles in their interaction networks (Table 2). The bird species that contributed the most to nestedness were *Hemixos castanonotus* (CN =1.663), *Parus venustulus* (CN =1.645), and *Pycnonotus sinensis* (CN =1.339), and the plant species that contributed the most to nestedness were *Alocasia macrorrhiza* (CN =2.891), *Cinnamomum camphora* (CN =2.236), and *Machilus nanmu* (CN =2.072; Table 2). None of the bird traits included in this study were significantly related to the network parameters (Table 3), and fruit color was the only plant trait that had a significant effect on species strength ($P = 0.013$; Table 4). The species strength of plants with black fruits ($2.69 \pm 1.25$) was significantly greater than that of plants with red fruits ($0.358 \pm 0.057$),

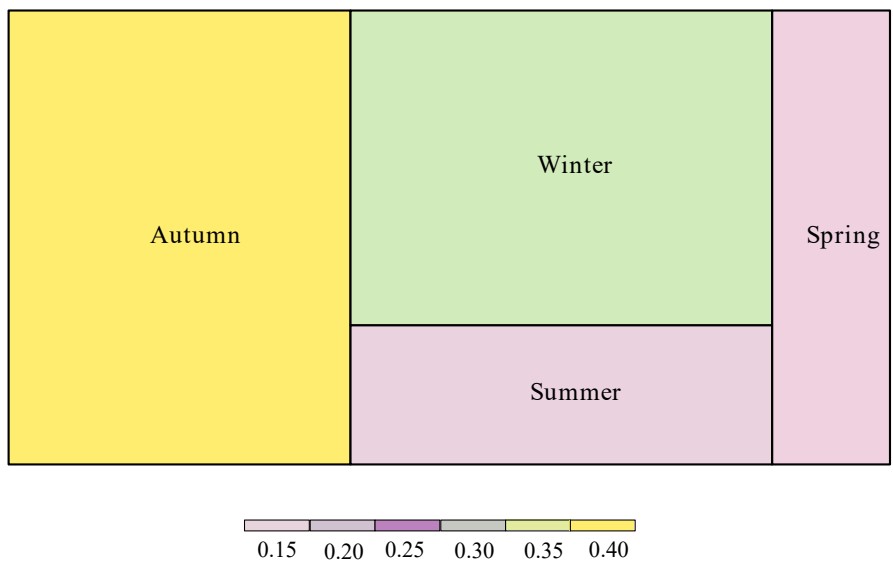

| 0.15 | 0.20 | 0.25 | 0.30 | 0.35 | 0.40 |

**Figure 4** Proportion of interaction connections in different seasons to the overall network.

**Table 1** Main quantitative description of seasonal variation of interaction network.

| Parameter | Autumm | Winter | Spring | Summer |
|---|---|---|---|---|
| Connectance ($z$-score) | −2.900 | −3.318 | −2.065 | −7.141 |
| weighted Nestedness ($z$-score) | −2.058 | −1.097 | −1.358 | −2.081 |
| Specialization ($z$-score) | 17.705 | 10.022 | 8.100 | 18.843 |
| Interaction diversity ($z$-score) | −17.704 | −10.022 | −8.100 | −18.843 |
| Interaction evenness ($z$-score) | −4.064 | −2.105 | −2.196 | −4.867 |
| Modularity ($z$-score) | 9.460 | 8.071 | 5.090 | 9.317 |

indicating that plants with black fruits occur more frequently and had greater species strength.

# DISCUSSION

Our results show that birds and fruit plants in urban green spaces can form a complex interaction network (Fig. 2). However, compared to the interaction networks of other urban green spaces (*Cruz et al., 2013*; *Zhang et al., 2022*), there were differences in bird and plant species in the interaction network of the Guilin Botanical Garden. This is likely related to its size, but also to the fact that the species composition of fruit plants in this Botanical Garden is relatively simple. Passerine birds were the main component of the observed interaction network (Table 2; Fig. 2), which may be due to their wide foraging for, the variety of fruits they forage, or their high adaptability to disturbed urban habitats. Thus, the presence of passerine birds with different body traits can improve the plant-frugivore networks of urban ecosystems.
**Table 2  Species-level metrics for the bird-plant frugivory interaction network in the Guilin Botanical Garden.**

| Bird species | Species degree | Species strength | Partner diversity | Effective partners | Specialization | CN |
|---|---|---|---|---|---|---|
| *Pycnonotus xanthorrhous* (Px) | 13 | 4.231 | 1.961 | 7.103 | 0.014 | 1.08 |
| *Zosterops japonicus* (Zj) | 9 | 1.839 | 1.767 | 5.851 | 0.156 | 0.994 |
| *Pycnonotus sinensis* (Ps) | 11 | 1.318 | 1.756 | 5.789 | 0.016 | 1.339 |
| *Hemixos castanonotus* (Hc) | 12 | 1.633 | 1.99 | 7.319 | 0.031 | 1.663 |
| *Ixos mcclellandii* (Im) | 13 | 1.495 | 1.842 | 6.309 | 0.044 | 1.149 |
| *Turdus merula* (Tm) | 8 | 0.544 | 1.511 | 4.531 | 0.08 | 1.214 |
| *Urocissa erythrorhyncha* (Ue) | 8 | 0.633 | 1.485 | 4.414 | 0.111 | 0.907 |
| *Parus major* (Pm) | 5 | 0.233 | 1.045 | 2.844 | 0.14 | 1.138 |
| *Pycnonotus jocosus* (Pj) | 7 | 0.295 | 1.703 | 5.491 | 0.072 | 0.558 |
| *Parus venustulus* (Pv) | 5 | 0.145 | 1.037 | 2.82 | 0.124 | 1.645 |
| *Streptopelia orientalis* (So) | 4 | 0.145 | 0.871 | 2.389 | 0.135 | 0.307 |
| *Copsychus saularis* (Cs) | 2 | 0.128 | 0.257 | 1.293 | 0.274 | 1.278 |
| *Turdus cardis* (Tc) | 7 | 0.25 | 1.758 | 5.801 | 0.126 | 0.809 |
| *Turdus hortulorum* (Th) | 5 | 0.112 | 1.378 | 3.966 | 0.097 | 1.184 |
| **Plant species** | | | | | | |
| *Cayratia japonica* (Cj) | 13 | 4.453 | 2.09 | 8.082 | 0.034 | 1.666 |
| *Cinnamomum camphora* (Cc) | 14 | 5.216 | 2.258 | 9.562 | 0.103 | 2.236 |
| *Osmanthus fragrans* (Of) | 7 | 0.738 | 1.557 | 4.745 | 0.128 | 1.488 |
| *Viburnum odoratissimum* (Vo) | 8 | 0.524 | 1.634 | 5.122 | 0.079 | 1.376 |
| *Machilus nanmu* (Mn) | 6 | 0.531 | 1.627 | 5.089 | 0.084 | 2.072 |
| *Ilex chinensis* (Ic) | 9 | 0.52 | 1.982 | 7.255 | 0.084 | 0.992 |
| *Magnolia grandiflora* (Mg) | 10 | 0.358 | 1.929 | 6.881 | 0.047 | 1.562 |
| *Bischofia javanica* (Bj) | 10 | 0.476 | 2.131 | 8.421 | 0.064 | 1.654 |
| *Rauvolfia verticillate* (Rv) | 7 | 0.323 | 1.68 | 5.367 | 0.111 | 0.581 |
| *Schefflera arboricola* (Sa) | 8 | 0.368 | 1.691 | 5.424 | 0.106 | 0.827 |
| *Tetrastigma planicaule* (Tp) | 7 | 0.224 | 1.759 | 5.805 | 0.162 | 0.378 |
| *Ilex cornuta* (Ico) | 8 | 0.239 | 1.786 | 5.963 | 0.104 | −0.347 |
| *Alocasia macrorrhiza* (Am) | 2 | 0.031 | 0.52 | 1.681 | 0.167 | 2.891 |

The observed network structure is less complex than that in the random network produced by the null model, probably because the intrinsic mathematical behavior of the null model generates more connected matrices than most observed networks (*Dormann et al., 2009*; *Costa, Silva & Ramos, 2016*). The degree of specialization and modularity of the observed interaction network was significantly higher than that of the null model. Highly specialized interactions are most often lost as a result of human intervention, and the extinction of one species may have fatal consequences for its interacting partners (*Sebastián-González et al., 2015*). A higher modularity in interaction networks is more common when bird dispersers have large morphological or ecological differences (*Donatti et al., 2011*).

We found seasonal variations in the characteristics of the interaction network (Fig. 4), which is consistent with our prediction (i). These differences could be attributed to

**Table 3  Results of generalized linear models (GLM) evaluating the effects of bird traits on their network roles.**

| Variable | Estimate | Standard error | t-value | P-value |
|---|---|---|---|---|
| **Species degree** | | | | |
| Intercept | 7.530 | 4.186 | 1.799 | 0.110 |
| Body mass | −0.023 | 0.024 | −0.974 | 0.359 |
| Body length | −0.007 | 0.027 | −0.257 | 0.804 |
| Wing length | 0.028 | 0.081 | 0.347 | 0.738 |
| Bill width | 0.284 | 0.387 | 0.735 | 0.483 |
| Wing loading | −18.880 | 23.318 | −0.810 | 0.442 |
| **Species strength** | | | | |
| Intercept | 2.185 | 1.541 | 1.418 | 0.194 |
| Body mass | −0.007 | 0.009 | −0.826 | 0.433 |
| Body length | 0.002 | 0.010 | 0.201 | 0.846 |
| Wing length | −0.009 | 0.030 | −0.295 | 0.775 |
| Bill width | −0.012 | 0.143 | −0.086 | 0.933 |
| Wing loading | −0.271 | 8.583 | −0.032 | 0.976 |
| **Partner diversity** | | | | |
| Intercept | 1.513 | 0.654 | 2.315 | 0.049 |
| Body mass | −0.003 | 0.004 | −0.805 | 0.444 |
| Body length | −0.001 | 0.004 | −0.200 | 0.847 |
| Wing length | 0.001 | 0.013 | 0.043 | 0.967 |
| Bill width | 0.035 | 0.060 | 0.585 | 0.575 |
| Wing loading | −1.416 | 3.641 | −0.389 | 0.707 |
| **Effective partners** | | | | |
| Intercept | 5.369 | 2.283 | 2.352 | 0.047 |
| Body mass | −0.012 | 0.013 | −0.935 | 0.377 |
| Body length | −0.006 | 0.015 | −0.389 | 0.708 |
| Wing length | −0.005 | 0.044 | −0.106 | 0.918 |
| Bill width | 0.172 | 0.211 | 0.813 | 0.440 |
| Wing loading | −4.350 | 12.718 | −0.342 | 0.741 |
| **Specialization** | | | | |
| Intercept | 0.122 | 0.101 | 1.214 | 0.259 |
| Body mass | 0.0003 | 0.001 | 0.433 | 0.677 |
| Body length | −0.0001 | 0.001 | −0.123 | 0.905 |
| Wing length | −0.001 | 0.002 | −0.416 | 0.688 |
| Bill width | −0.001 | 0.009 | −0.054 | 0.959 |
| Wing loading | 0.311 | 0.561 | 0.554 | 0.594 |

the seasonal decline of other food sources, such as insects, and the arrival of wintering populations, which intensifies competition for food resources (*Cruz et al., 2013*; *Yang, Albert & Carlo, 2013*). The number of observed frugivorous bird and ripe fruit species was significantly associated with interaction connections in all four seasons, which is in agreement with previous studies that have shown that an increase in fruit abundance significantly improves bird richness and network complexity (*Ramos-Robles, Andresen &*

**Table 4** Results of generalized linear models (GLM) evaluating the effects of plant traits on their network roles.

| Variable | Estimate | Standard error | *t*-value | *P*-value |
|---|---|---|---|---|
| **Species degree** | | | | |
| Intercept | 9.901 | 6.710 | 1.475 | 0.184 |
| Fruit mass | 0.146 | 5.579 | 0.026 | 0.980 |
| Fruit length | −0.710 | 0.977 | −0.727 | 0.491 |
| Fruit diameter | 1.542 | 1.117 | 1.380 | 0.210 |
| Fruit volume | −0.003 | 0.009 | −0.305 | 0.770 |
| Fruit color | −3.365 | 2.035 | −1.654 | 0.142 |
| **Species strength** | | | | |
| Intercept | 2.422 | 3.013 | 0.804 | 0.448 |
| Fruit mass | 0.948 | 2.505 | 0.378 | 0.717 |
| Fruit length | 0.231 | 0.439 | 0.526 | 0.615 |
| Fruit diameter | 0.353 | 0.502 | 0.703 | 0.505 |
| Fruit volume | −0.003 | 0.004 | −0.732 | 0.488 |
| Fruit color | −3.036 | 0.914 | −3.322 | 0.013 |
| **Partner diversity** | | | | |
| Intercept | 1.590 | 1.066 | 1.491 | 0.179 |
| Fruit mass | 0.090 | 0.886 | 0.101 | 0.922 |
| Fruit length | −0.189 | 0.155 | −1.215 | 0.264 |
| Fruit diameter | 0.292 | 0.178 | 1.645 | 0.144 |
| Fruit volume | −0.0004 | 0.001 | −0.256 | 0.805 |
| Fruit color | −0.168 | 0.323 | −0.521 | 0.619 |
| **Effective partners** | | | | |
| Intercept | 4.151 | 4.517 | 0.919 | 0.389 |
| Fruit mass | 1.390 | 3.755 | 0.370 | 0.722 |
| Fruit length | −0.605 | 0.658 | −0.921 | 0.388 |
| Fruit diameter | 1.383 | 0.752 | 1.839 | 0.109 |
| Fruit volume | −0.004 | 0.006 | −0.639 | 0.543 |
| Fruit color | −1.274 | 1.370 | −0.930 | 0.383 |
| **Specialization** | | | | |
| Intercept | 0.074 | 0.093 | 0.792 | 0.454 |
| Fruit mass | 0.048 | 0.077 | 0.624 | 0.552 |
| Fruit length | 0.018 | 0.014 | 1.313 | 0.231 |
| Fruit diameter | −0.018 | 0.016 | −1.138 | 0.293 |
| Fruit volume | −0.00004 | 0.0001 | −0.326 | 0.754 |
| Fruit color | −0.005 | 0.028 | −0.178 | 0.863 |

*Díaz-Castelazo, 2016*; *Schneiberg et al., 2020*). The winter networks had higher weighted nestedness ($z$-score) and interaction evenness ($z$-score; Table 1), suggesting more effective resource utilization. Nevertheless, the structure of the network in winter was determined by a small number of high-frequency interactions, especially with *Cayratia japonica* and *Cinnamomum camphora*. The higher specialization ($z$-score) observed in summer may be related to birds' response to low plant diversity (*Hernández-Montero et al., 2015*). Low

plant diversity reduces the availability of fruit resources, causing the impoverishment of frugivorous diversity and bird-fruit interactions (*Menke, Böhning-Gaese & Schleuning, 2012*). In addition, specialization is also related with resource complementarity (*Silva et al., 2016*). A high level of specialization indicates a high degree of niche differentiation (*Blüthgen, 2010*; *Sebastián-González et al., 2015*) and decreased competition, facilitating species coexistence (*Silva et al., 2016*).

Our analysis showed that of the 14 bird species recorded, the three most important to the interaction network were *Hemixos castanonotus*, *Parus venustulus*, and *Pycnonotus sinensis* based on their contribution to nestedness (Table 2). These results indicate that these bird species play an important role in maintaining the stability of the network structure because they interact with most of the same plants that other birds in the network use (*Sebastián-González, 2017*). These same species are among the most important plant seed dispersers in urban green spaces (*Zhang et al., 2022*), because they provide the food resources, perching sites, and protection that these birds require. Additionally, these birds are resident species in urban green spaces and are present in large numbers; therefore, the stability of urban ecosystems may depend on their seed dispersal capabilities.

No significant association was observed between species traits and network roles, except for the effect of fruit color on species strength (Tables 3; 4). This is contrary to the results of previous studies (*Saavedra et al., 2014*; *Pigot et al., 2016*) and is inconsistent with our prediction (ii). These differences can likely be explained by the following two factors: the contribution of some species to the network was significantly higher than that of other species, masking the impact of species traits (*Costa, Silva & Ramos, 2016*), and the weighted analyses in this study had small sample sizes that were based on only one year of research data. Plants with black fruits had greater species strength, which could be because black fruits are especially conspicuous against natural backgrounds and the appear more frequently in this region (*Duan, Goodale & Quan, 2014*; *Zhang et al., 2022*). There is also a significant positive correlation between lipid nutrients and color; therefore, color may be a signal of seed maturity and nutritional content (*Schaefer, Valido & Jordano, 2014*). Finally, evergreen fruit plants, such as *Cinnamomum camphora* and *Bischofia javanica*, can provide temporary shelter when birds forage in urban habitats; therefore, birds may choose to forage the fruits of these species for survival.

## CONCLUSIONS

Our results indicate significant seasonal differences in the structure and characteristics of the interaction network between plants and frugivorous birds in urban green spaces. None of the plant or bird traits were significantly correlated with the functional roles of the species in the network structure, except for the effect of fruit color on species strength. The sample size and study area may be possible reasons for the absence of other correlations. Therefore, in future research, a large number of field observations and both quantitative and qualitative analyses are necessary to help us understand the role of urban green space in maintaining species diversity and ecosystem functions.

## ACKNOWLEDGEMENTS

We would like to thank the staff of the Guilin Botanical Garden for their assistance during our fieldwork.

### Funding

The National Natural Science Foundation of China (No. 32170492; 32270504), the Guangxi Natural Science Foundation (No. 2019GXNSFD-A245021), the Scientific Research Foundation of Guangxi Normal University for Nationalities (No. 2021BS002), the fourth batch of characteristic discipline construction projects in Ethnic Colleges and universities approved by the Department of Education of Guangxi Zhuang Autonomous Region (Ethnic Ecology). Funding was also provided by the Key Laboratory of Ecology of Rare and Endangered Species and Environmental Protection (Guangxi Normal University), Ministry of Education, China, and the basic ability enhancement program for Young and Middle-Aged Teachers Of Guangxi (No. 2020KY20017). The funders had no role in study design, data collection and analysis, decision to publish, or preparation of the manuscript.

### Grant Disclosures

The following grant information was disclosed by the authors:
National Natural Science Foundation of China: 32170492, 32270504.
Guangxi Natural Science Foundation: 2019GXNSFD-A245021.
Scientific Research Foundation of Guangxi Normal University for Nationalities: 2021BS002.
Key Laboratory of Ecology of Rare and Endangered Species and Environmental Protection (Guangxi Normal University), Ministry of Education, China.
The fourth batch of characteristic discipline construction projects in Ethnic Colleges and universities approved by the Department of Education of Guangxi Zhuang Autonomous Region (Ethnic Ecology).
The basic ability enhancement program for Young and Middle-Aged Teachers of Guangxi: 2020KY20017.

### Competing Interests

The authors declare there are no competing interests.

### Author Contributions

- Guohai Wang conceived and designed the experiments, performed the experiments, prepared figures and/or tables, authored or reviewed drafts of the article, and approved the final draft.
- Yang Huang conceived and designed the experiments, performed the experiments, prepared figures and/or tables, and approved the final draft.
- Wei Yao conceived and designed the experiments, performed the experiments, prepared figures and/or tables, and approved the final draft.
- Qiuchan Huang analyzed the data, prepared figures and/or tables, and approved the final draft.

- Yongping Huang analyzed the data, prepared figures and/or tables, and approved the final draft.
- Lijuan Wei analyzed the data, prepared figures and/or tables, authored or reviewed drafts of the article, and approved the final draft.
- Qihai Zhou performed the experiments, analyzed the data, authored or reviewed drafts of the article, and approved the final draft.

## Data Availability

The raw measurements are available in the Supplementary File.

## Supplemental Information

Supplemental information for this article can be found online at http://dx.doi.org/10.7717/peerj.15028#supplemental-information.

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
