# Peer review of "Structure and characteristics of the plant-frugivore bird network from the Guilin Botanical Garden"

_PeerJ, doi:10.7717/peerj.15028_

## Round 0.1 · original submission · Major Revisions

Kindly address the key issue highlighted by both of the reviewers

Reviewer 1 ·

Basic reporting

The manuscript entitled “Structure and characteristics of plant-frugivore
network in Guilin Botanical Garden” analyzed plant-bird frugivore network structure and species roles in one year of survey.
After reading the manuscript, I think that the main aim of the study is not clearly identified and expressed in the text. Moreover, I have some general questions about the study that I suggest being revised and answered in another version of the manuscript: What is the main motivation for conducting this study? Why were authors interested in evaluating this type of network in an urban area? Which are the main impacts of this study? What we already know about urban ecological networks? Why were authors interested in evaluating species roles?

Following, I include my specific comments to the manuscript.
Abstract
L14: I would suggest expressing “ecosystem functions and diversity in communities”
L16: I would suggest expressing “among plant and frugivore bird species”
L20: I would suggest expressing “Spring and summer networks”


Introduction
L33: is a taxonomic group missing in the phrase “the interaction of frugivorous and…”? please correct the sentence.
L34: please clarify the expression “is a crucial ecological process that connects the successive plant generations”
L77-78: please clarify the second question, it is confusing “plant and frugivorous bird
function role in the interaction networks”
L79: what is the meaning of “more complicated networks”? please clarify the term. Moreover, please, specify in the Introduction the type of role you evaluated for plants and birds, either functional or topological. Up to here, readers don’t understand what you are referring by “role”, there is no information about species roles in the Introduction. Please clarify this point in the text.

Experimental design

Methods
L103-104: please correct the sentence “The observation frequency at least 8d observations per month”, a verb is missing for understanding the sentence.
L141: please clarify the type of null model you used, and statistic toll employed.

L149: again, the concept of network roles needs to be more clearly defined in the manuscript. Are you evaluating functional roles for plant and bird species? Why are you interested in evaluating the relationship between species traits and network roles? Theoretic information about these concepts is missing in the text.

Validity of the findings

Results

I would suggest including information about species origin, were plant and bird species mainly exotic in this system? What would you expect about species origin in the type of system evaluated?

L201-202: please check the sentence, it has some mistakes.


Discussion
L223: why are you assuming that the study area evaluated has a “limited amount of biodiversity”? some green urban areas are highly diverse and conserved. Is the study area highly fragmented or is under severe disturbance? If this is the case, you need to include the information in Methods.
As you are not comparing among urban areas o between urban vs natural areas, it is difficult to understand the results and to compare them with other situations.
L224: what do you mean by “Passerine birds occupied a dominant position in the whole interaction network”?
L234-236: this sentence needs to be revised and re written, it is complicated to understand its meaning.

Conclusions
L279: in a future, to evaluate the effect of urbanization on urban networks I would suggest defining a different experimental/survey design.

Reviewer 2 ·

Basic reporting

The manuscript by Wang et al evaluates the bird-plant frugivore network in an urban botanical garden. The manuscript is easy to read and understand, and the introduction provides a good background on thematic. Nonetheless the references used during the material and methods, are not always the most adequate, and this should be addressed/corrected. Please see my detailed comments. Some of the figures should also be improved.

Experimental design

The sampling methodology seems good and a lot of effort seems to have been placed in collecting the data. Nonetheless, there are some details, namely sampling effort that should be a little more specific in the material and methods section. I also have some concerns regarding the statistical analysis. While some network metrics used should be explained a little better, my main concern is related with the use of simple correlations instead of more robust approaches like generalized linear models. Once again, please see my detailed comments.

Validity of the findings

I do not disagree with the authors results and findings, but I think they should be somehow better supported with other analysis.

Additional comments

Detailed comments
Line 34 – please remove the “.” before the reference.
Line 79 – You mean “complex” instead of “complicated”.
Line 83 – Can you please check the coordinates presented? When I searched on google maps, it does not go the Guilin botanical garden. Usually, latitude is also presented before the longitude.
Lines 103-104 – It was performed one day per transect, with a minimum of 2 visits per transect in each month? Can you please be a little bit more specific where? This is important to have an idea of the sampling effort.
Line 127- I suggest to change the “network-level” by “networklevel” that is the how the function is called.
Lines 127-128 – You are using the R reference for the bipartite package. Please change this.
Line 128 – maybe a better term for “statistics” is “network metrics”.
Line 131 – Cruz et al. 2013 is not the best reference for connectance, as it is simply one of many studies that uses this metric. Consider replacing it for a reference that uses connectance in an ecological network context, for example. In the next text line, including species level metrics, are similar cases, that you may consider replacing the metric reference.
Also, what was the nestedness measure used? Is the “nestedness temperature of the matrix”. If it is, can you briefly explain why you used it instead of the more commonly used (W)NODF?
Line 140 – Maybe you can also calculate the network modularity.
Lines 141 – Please specify the null model used and why it was chosen.
Line 162 – Change “species-level” by “specieslevel”.
The bipartite reference is also incorrect.
Lines 164-168 – This statistical approach is not very common nowadays. Why did you not use linear models (e.g., GLMs) where you can test all the traits together and adjust the distribution to the network metric distribution? Some of the variables tested are likely highly correlated, and therefore the results obtained should be very similar. You should consider redoing these analyses or explain better your choice.
Line 174 – Consider rephrasing this sentence, avoiding to start the paragraph with a number.
Lines 183-191 – This is the Fig. 2 caption? It seems odd that it is presented in the middle of the results.
Lines 203-206 – Consider using z-scores to compare network metrics between seasons, instead of comparing their raw values. Network size is known to affect some network metrics and their size is considerably different, especially between autumn and winter vs spring and summer.
Line 211 – As I previously mentioned, I do not think this is the best statistical approach for your question.
Line 221 – This is a description of your results. Please consider the first part of this sentence.
Figure 2 – Please consider changing the colours, as these are not very colour-blind friendly.

---

## Round 0.2 · Major Revisions

Kindly check the comments of reviewers and make the changes.

Reviewer 1 ·

Basic reporting

The main motivation and impact of the study need to be expressed in the Introduction. Authors answered these questions in the rebuttal letter; however, this information is missing in the manuscript.

I would suggest including some brief description of the origin of plants and bird’s species in the Garden (native-exotic species), whether plants were planted in this site, whether is an old park etc, a brief history of the Garden that helps to understand the usage history of the site. This will be helpful to interpret the results of this study and to applicate and compare it with other studies in different study sites.

The results found here expressed one year of sample, this may be a limitation of the results. Authors mentioned this in the conclusion; however, I would suggest including a mention of this in the Discussion section.

Introduction

L61: please clarify what do you mean by “not all the plants or birds in the network are equally important”. The term “important” is confusing, do you mean that some species concentrate most interactions in the network? So, “important” according to the number of interactions. Please briefly clarify this in the sentence.

Experimental design

L104: please check the spelling of the following sentence, it contains some language mistakes “Once birds identified foraging for plant fruits, the species of both the birds and the plants foraged, as well as the number of fruits, the number of birds per visit, and the foraging time, were all recorded”.
L111: please check the spelling “when birds were found be to feeding on plant fruits”

Data analysis
L165: please correct “specieslevel”
L177: here you introduce the term “stability of the network”, I would suggest including a brief explanation of this term.

Validity of the findings

Conclusions

The conclusions section stated that “Therefore, future studies should expand the research area and adopt a different experimental design to evaluate the effect of urbanization on urban frugivorous bird and plant interaction networks.”, why you suggest different experimental design would be important for evaluating effect of urbanization? This needs to be previously discussed, moreover, the main purpose of this study is not related with evaluating the effect of urbanization on networks. I would suggest including the main aim of the study in the Introduction, and clearly expressed the main impacts of the results found to understand how green urban spaces may help to maintain species interaction networks and ecosystem functionality.

L298: please check the spelling “The study sample size or the limited geographic area included in the study may be possible reasons other correlations were not observed.”

Additional comments

no additional comments. All my specific comments were listed in previous comments boxes.

Reviewer 2 ·

Basic reporting

The overall structure of the manuscript is good, as well as manuscript readability. The authors corrected several references as previously suggested and I think the current manuscript version is considerably better regarding this subject. However, I think it can still be slightly improved (please see the “additional comments” section for more details).

Experimental design

I have no further considerations regarding the experimental design.

Validity of the findings

Overall, I think the analysis performed is more robust than in the previous manuscript version, but I still have some notes regarding the analysis performed. For example, you should consider using the weighted NODF, instead of its unweighted version. The wNODF is usually less biased to different sampling completeness levels. The use of z-scores can also provide a useful way to compare the different network metrics across seasons (it is calculated as (observed−mean(null))/sd(null)).

Additional comments

The number of the lines indicated is from the pdf file.
Lines 40 and 41 – Consider adding a reference at the end of this sentence to support it.
Lines 68 and 69 – Consider adding a reference at the end of this sentence to support it.
Lines 77 and 78 – Many botanical gardens around the world, due to the higher plant diversity, have fleshy fruits throughout the year, or at least during a larger proportion of the year in comparison with the native flora, specially outside tropical and sub-tropical climates. Maybe you can mention this and use Guilin botanical Garden as a case study of something that also happens in many other botanical gardens around the world, to given a more global context to your work.
Lines 106 and 107 – This information is repeated on line 111. Please avoid having the same information repeated, and remove it from these lines.
Line 121 – Have you considered using if a plant species is native or exotic? Many botanical gardens have a considerable number of exotic plants. Is this not the case of the Guilin botanical Garden? If there are exotic and native plants this can be an important variable with considerable conservation implications that I think should/could be addressed (if there are exotic and native plants in your network).
Line 206 – These are ‘random’ networks.
Line 216 – From where are these R2 values? I may be missing something, but I do not find these correlations mentioned in the data analysis section.
Line 228 – You cannot be sure if black fruits attract more birds or not. They can attract the same number of birds but be more common and therefore have a higher species strength. Please consider rephrasing this sentence.

---

## Round 0.3 · Minor Revisions

Dear editor:
The manuscript is improved based on reviewer comments, there are some minor which can be edited before print. I suggest minor revisions.
Thank you

Reviewer 1 ·

Basic reporting

The authors have considered all the comments and suggestions made in previous revisions of the manuscript. It has improved in quality. The impacts and likely impacts of the resuls were included in the manuscript, also the limitations of the study are now discussed in the text.

Experimental design

-

Validity of the findings

-

Additional comments

-

Reviewer 2 ·

Basic reporting

I still have a few corrections and doubts that the authors need to consider (please see additional comments sections)

Experimental design

I have no further considerations regarding the experimental design.

Validity of the findings

I have no further considerations regarding the experimental design.

Additional comments

The number of the lines indicated is from the pdf file.
Line 14 – Consider ending the sentence here (replacing “;” by a “.”).
Line 25 – I do not think you need to mention “(N=1,000) produced by the null model” in the abstract. To me these are details that should be only in the methods section.
Line 32 – Consider changing the keywords, as they should not include words present in the manuscript title.
Lines 107-108 – “Safari l0 × 42 zoom binoculars” Please check the binoculars. The first number usually represents the amplification and the second the diameter of the lens. You say ‘zoom binoculars but the amplification is only a number, usually zoom binoculars are for example 10-30 × 42.
Line 122 – Please remove ‘are’.
Line 191 – Is this ‘contribution’ the CN?
Lines 203 and 204 – Can you check this sentence; it may need some rewriting? You mean that 92.89% of the bird species forage on more than 5 plant species? (This is what is possible to see on Fig.2). If not, how can you know that a bird (individual) forage on more than one individual plant, when the birds are not marked? Were you able to visually follow most individual birds (92.89%) and see them feeding in 5 or more plants?
Lines 227 to 229 – I still do not understand what is this ‘contribution’ you are relating with the number of birds and plants. This should be better explained in the methods. I am also guessing that the first R value is for birds and the second for plants (but you should be clear on the text), and the second correlation is not significant so you should say that it is positive but not significant.
Line 241 – Something seems to be missing here. You mean more commonly eaten?
Line 248 – Why don’t you compare with Cruz et al. 2013 that you mention in the introduction?
Line 257 and 258 – You already have the method/algorithm used for each network metric in the methods. I do not think it is necessary to repeat them here (just mention the metric).
Lines 256 to 259 – Here you are describing results and repeating the previous section. Consider rephrasing, shortening or even removing these lines.
Lines 279 and 280 – How will a low plant diversity be responsible for a higher specialization? Can you explain this a little better?
Line 297 and line 298– Delete ‘individual’.
Figure 2 – Change B1, B2 etc and P1, P2, etc for abbreviations of the bird and plant names, respectively.

---

## Round 0.4 · accepted · Accept

The manuscript is improved based on reviewer comments, my suggestion is to accept it now that you have made the necessary changes.